# The Archipelago of Cultural and Creative Industries: A Case Study of the Basque Country

**Xabier Barandiaran-Irastorza** [1] , **Simón Peña-Fernández** [2,*] and **Alfonso Unceta-Satrústegui** [2]

[1] Faculty of Social and Human Sciences, University of Deusto, 20012 San Sebastian, Spain; xabier.barandiaran@deusto.es

[2] Faculty of Social and Communication Sciences, University of the Basque Country (UPV/EHU), Barrio Sarriena, 48940 Leioa, Spain; alfonso.unceta@ehu.eus

[*] Correspondence: simon.pena@ehu.eus

**Abstract:** The cultural and creative industries (e.g., digital/audiovisual content, publishing and print media, etc.) constitute an emerging business sector focused on aligning cultural and creative production with profitability and market criteria that encompasses a diverse range of entities, generates employment, boosts GDP (Gross Domestic Product) and drives innovation. This paper analyses the impact of the cultural and creative industries on the economy of the Basque Country and provides information concerning the number of companies present in this sector, their legal structures, annual turnovers, size in terms of the number of people they employ and relative ability to secure public funding for entrepreneurial support and creative projects. Our findings indicate that businesses of this nature in the Basque Country form an ecosystem similar to an archipelago in which companies with a strong entrepreneurial focus, high annual turnovers and a national and international clientele coexist alongside smaller, less profit-oriented organisations devoted to regional cultural development.

**Keywords:** public policy; creativity; cultural activities; innovation; cultural and creative industries; regional development; Basque Country; A14; R58

## 1. Introduction

The interchangeable use of the terms culture and creativity (Cunningham 2002) to refer to a wide range of entities and activities that may or may not be commercially oriented or carried out within business frameworks has generated a vigorous debate among sector players and researchers working in the area of culture and creativity (Galloway and Dunlop 2007; Hesmondhalgh 2008; Bustamante 2009; O'Connor 2010).

While much of the creative and cultural work being done today clearly does not respond to market pressures or criteria, a significant amount is also being produced in structured corporate environments (Howkins 2007). In light of the emergence of the latter scenario, culture may now be also considered an economic sector that industrializes both demand and the means of satisfying the needs it creates (Maiso 2011).

Although many reject the idea of considering culture from an economic perspective, as numerous international organisations are quick to point out, commercially oriented cultural and creative ventures generate economic value (UNCTAD 2008; UNESCO 2009; Annoni and Kozovska 2010; TERA Consultants 2014; Chapain and Stryjakiewicz 2017). The European Commission's green paper on the potential of cultural and creative industries stresses the important contributions made by small enterprises in the areas of creativity and innovation, underscoring the decisive role they play in 'scouting for new talent, developing new trends and designing new aesthetics' (European Commission

2010). Viewed from a broader perspective, cultural and creative industries also contribute to economic growth and generate new employment opportunities (Higgs et al. 2008; Power 2011).

It is therefore not surprising that cultural and creative industries have made their way onto public policy agendas. The European Commission, for example, has highlighted the role of the creative sector at both the national and transnational level in a number of its initiatives and integrated it into its Europe 2020 strategy, as much for its potential as a driver of innovation and growing economic weight as for its social and cultural significance (European Commission 2011).

Despite these industries' importance, their activity is difficult to analyse due to the wide variety of businesses they encompass and a longstanding perception that their economic contribution is relatively marginal (Bonilla et al. 2012). The fact that cultural and creative endeavours were seldom conceived as profit-making ventures until fairly recently, combined with these industries' lack of a strong business culture and limited knowledge of management techniques, market characteristics and business models, makes any attempt to assess their activity even more difficult (Bedoya 2016).

In short, any analysis of the cultural and creative industries undertaken during the last two decades has faced three types of challenges. The first is an ongoing, unresolved debate regarding their nature kept alive by those who continue to resist considering them in commercial terms (Flew and Cunningham 2010). The second is the lack of a common, internationally accepted system for classifying their activities clearly into one camp or another, which has given rise to endless disagreement concerning which types of professionals, activities and end products belong to each sector (Boix and Lazzeretti 2012). The third (and consequence of the second) is the lack of comparable statistical registers, which complicates the task of analysing a sector comprising numerous micro-enterprises enormously and reduces the possibility of obtaining comparable data by means of standard indicators (Pfenniger 2004).

Despite the fact that there are numerous recent European studies on the cultural and creative industries at European level (Boschma and Fritsch 2009; Piergiovanni et al. 2012; Thomas et al. 2013; Andres and Chapain 2013; Boix et al. 2016; Lazzeretti et al. 2016) research conducted in Spain to date has been limited in scope (CEIN 2005; Casani 2010; Santos 2011; Boix and Lazzeretti 2012; Rodríguez et al. 2017; Verón-Lassa et al. 2017; Murciano and González 2018). At the local level, there are also several monographic and comparative studies related to smart and creative cities (Bakici et al. 2012; Méndez et al. 2012; Guerrero and Navarro 2012). In the case of the Basque Country, where apart from a study produced by KEA European Affairs (2008) on the province of Biscay, only superficial attention has been paid to this topic, no precise information has been compiled regarding the size, structure and organisational nature of companies engaged in this kind of activity. The Basque Autonomous Community (2,173,210 inhabitants) is located in northern Spain and is divided into the Historical Territories of Bizkaia (1,141,442 inhabitants), Alava (321,777 inhabitants) and Gipuzkoa (709,991 inhabitants). Each of these territories has its own provincial government, their Provincial Councils and Regional Laws, with broad powers for the administration and socio-economic and political management of each region.

However, at present the subject continues creating a great interest in the public administrations. Thus, Basque Government's Euskadi 2020 Science, Technology and Innovation Plan (Basque Government 2014) applies the Research and Innovation Smart Specialization Strategy (RIS3), and identifies leisure, entertainment and culture as one of the opportunity niches linked to the territory. Within this area, the plan specifically mentions the importance of developing cultural and creative industries such as language industries, videogames, etc. Likewise, based on all the above, the Provincial Government of Gipuzkoa has incorporated among its experimental projects within the framework of the Etorkizuna Eraikiz initiative (Building the Future) a Laboratory for the Promotion and Strengthening of Cultural and Creative Industries.

In summary, if globally, the analysis of data on cultural and creative industries in the economy shows its undoubted impact and its treatment as a strategic sector by public administrations (Aguado 2010), existing studies show the potential for creative industries to grow a regional economy

(Fleischmann et al. 2017b; Wu and Li 2018), and their geographic clustering has centred much research and policy attention (Fleischmann et al. 2017a).

In this context, this paper aims to describe the characteristics of cultural and creative industries in the Basque Autonomous Community, measure their economic impact and the effects of the financing they receive from the different public administrations.

## 2. Materials and Methods

In light of the background information provided above, the overall purpose of this study has been to piece together an accurate picture of the size, characteristics and economic impact of the cultural and creative industries based in the Basque Country. In pursuit of this goal, we defined the following four specific objectives:

O1—Determine the number of enterprises pertaining to the cultural and creative industries in the Basque Country.

O2—easure the impact of the Basque Country's cultural and creative industries on the Basque economy.

O3—Estimate the number of jobs directly generated by the Basque Country's cultural and creative industries.

O4—Analyse public sector programmes in the Basque Country that provide financial support for the region's cultural and creative industries and the determine the extent to which they benefit different types of Basque businesses falling into this category.

Once these objectives had been established and a review of the existing literature conducted, we formulated the following research hypotheses:

**Hypotheses 1 (H1).** *The cultural and creative industries based in the Basque Country have a significant, measureable economic impact in terms of their size, productivity and employment opportunities they generate.*

**Hypotheses 2 (H2)**. *The Basque Country's cultural and creative industries are diverse, in the sense that all of the sectors pertaining to them produce identifiable and measurable activity, but the distribution of business activity is uneven with the greater part corresponding to a small number of players.*

**Hypotheses 3 (H3).** *Cultural and creative industries in the Basque Country take advantage of government funding programmes supporting entrepreneurship and creativity administered at various levels throughout the autonomous community.*

In terms of available statistical information, the vast majority of studies related to business topics published to date have not drawn distinctions between companies in the cultural and creative industry and those in other sectors. Moreover, reference codes used to classify cultural and creative industries into sectors do not always adequately reflect the activities they actually engage in. Although most classification codes used in statistical reports issued by the British government's Department for Digital, Culture, Media and Sport (DCMS 1998), the World Intellectual Property Organization (WIPO 2003), and UNCTAD (2008) coincide, they differ in certain respects.

To classify the cultural and creative industries analysed in this study, we divided applicable codes provided in the Statistical classification of economic activities in the European Community (NACE, Rev. 2, EUROSTAT 2008) (2008) and transposed in Spain as the Clasificación Nacional de Actividades Económicas (CNAE-09) (INE 2008). Spain's national classification of economic activities (CNAE-2009) divides certain NACE Rev. 2 classifications into smaller categories. For instance, NACE Rev. 2 category 59.11 (Motion picture, video and television programme activities) is divided in the CNAE-2009 registry into the more specific categories 59.15 (Motion picture and video production activities) and 59.16 (Television programme production activities). The CNAE system likewise divides NACE Rev. 2 category 59.13 (Motion picture, video and television programme distribution activities) into the smaller categories 59.17 (Motion picture and video distribution activities) and 58.18 (Television distribution activities).

This classification has been adapted into the 15 sectors (Table 1) established in 2014 by the Basque Government to classify businesses pertaining the cultural and creative industries. Gastronomy is included in the list, as it is defined as an opportunity niche in the RIS3 and ICC strategy of the Basque Government. As no code contained in the CNAE-09 registry corresponded to handcraft activities, no data could be compiled or analysed for this sector.

**Table 1.** Cultural and creative industry sectors and corresponding CNAE-09/NACE Rev. 2 codes.

| Sector | CNAE-09/NACE Rev. 2 Codes |
| --- | --- |
| Audiovisual content (1) | 5912, 5914, 5915, 5916 5917, 5918, 6010, 6020 |
| Digital content (1) | 5829, 6311, 6312, 6209 |
| Design (1) | 7410 |
| Architecture (2) | 7111 |
| Fashion (2) | 1320, 1330, 1391, 1392 1399, 1411,1412, 1413 1414, 1431, 1439,1520 3212, 3213, 1512, 1420 |
| Video games (1) | 5821 |
| Advertising and Marketing (2) | 7311, 7312, 7021 |
| Handcrafts (2) | - |
| Performing arts (1) | 9001, 9002, 9003, 9004 |
| Visual arts (1) | 7420 |
| Music (1) | 5920, 3220 |
| Gastronomy (2) | 5610 |
| Language services (2) | 7430, 8559 |
| Publishing and print media (2) | 1811, 1812, 1813, 1814 5811, 5813, 5814, 5819 1820 |
| Cultural heritage (1) | 8552, 9102, 9103, 9105 9106 |

(1) Sector completely analysed; (2) Sector partially analysed.

The Iberian Balance Sheet Analysis System database (2019), which provides financial information on more than three million enterprises in Spain and Portugal, was used to identify companies pertaining to the cultural and creative industries in the Basque Country and measure their economic activity. Information contained in this system managed jointly by Informa D&B and Bureau Van Dijk is updated monthly in accordance with AENOR ISO 9001:2008 standards.

Due to the complexity of verifying that CNAE codes assigned to individual companies reflected the activities that each was actually engaged in, it was only possible to analyse 8 sectors completely and another 6 partially during this phase of the study. The remaining sector, defined by the Basque government as handcrafts, could not be analysed at this point.

In light of the role public funding plays in enterprise development, we analysed the distribution of government funding disbursed via 41 entrepreneurial support grant competitions administered by the Basque regional government and the provincial governments of Gipuzkoa and Biscay during 2014 and 2015.

We likewise took a close look at public funding available in the Basque Country for creativity, which includes grants for work in specific disciplines such as dance, music and the performing and visual arts as well as more general projects. A thorough analysis was made of the distribution of funding awarded in this area via 82 public grant competitions administered by the Basque regional government and its three provincial governments during the period 2014–2015.

## 3. Results

### 3.1. A Numerical Breakdown of the Cultural and Creative Sector in the Basque Country

An analysis of data extracted from the SABI database (SABI 2018) indicates there are currently 5071 cultural and creative enterprises in the Basque Country (Figure 1). The language industries account for more companies (1227 or 24%) than any other sector in the industry, followed by publishing and print media (832 or 16%), advertising and marketing (721 or 14%) and architecture (639 or 13%). Two out of three companies in this autonomous community pertain to these four sectors.

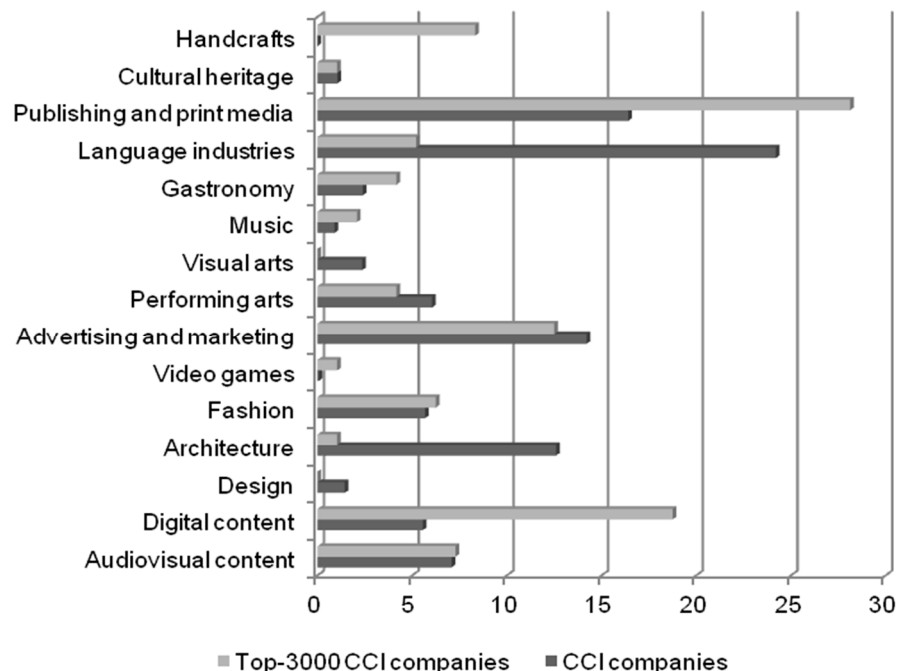

**Figure 1.** Relative weight of sectors within the Basque cultural and creative industries and relative presence of enterprises in each sector on the list of Basque Country's top 3000 companies (in %). Source: Compiled by the authors on the basis of SABI and Empresa XXI data.

On the other end of the spectrum, companies involved in video games (3 or 0.06%), music (46 or 0.9%), cultural heritage (53 or 1%) and design (73 or 1.4%) constituted the four smallest sectors in the sample.

In terms of geographic distribution, 53% of the enterprises examined for this study were located in the province of Biscay, 32% in the province of Gipuzkoa and 15% in the province of Álava.

If, on the other hand, we focus on the relevance of these enterprises rather than their total number, we can refine our portrait of the Basque cultural and creative industries by identifying those that figure among the companies with the greatest impact on the Basque economy. The ¿Quién es quién en la economía vasca? report (Empresa XXI 2014) contained relevant information on this point.

Of the 3087 companies mentioned in this report, which ranks firms on the basis of sales volume, 96 (3.11%) were classified as cultural and creative industry enterprises.

Thirteen of the fifteen sectors associated with the cultural and creative industries are represented on Empresa XII's list of the Basque Country's 3000 top performing companies. The three that figure the most prominently are publishing and print media (27 companies), digital content (18) and advertising and marketing (12). In contrast, no company devoted to design or visual arts and only one in the areas of architecture and cultural heritage made the list.

A comparison of the total number of companies in the Basque cultural and creative industry sector and the percentage of them financially successful enough to contribute substantially to the Basque economy helps us to visualize the archipelago they constitute and identify the top-performing sectors within the overall group. Findings indicate that 45 of the 96 cultural and creative industry companies on the 2014 top 3000 list correspond to the areas of publishing and print media and digital content. Basque public radio and television network EITB and the media conglomerate Vocento were among those in their categories that made the grade.

On the other hand, no firm in dynamic sectors with a crowded field of players such as the language industry (the sector with the highest number of companies, includes translation, content-terminology, lexicography, etc.—language teaching and language technologies) or architecture (the fourth-largest sector from the same perspective) ranked among the Basque Country's top 3000 companies.

The majority of cultural and creative industry enterprises based in the Basque Country (57%) are private limited companies. Non-profit associations account for another 14%. The picture nevertheless varies from one sector to another. Whereas most of the enterprises in the digital content, video games and gastronomy sectors are private limited companies, a significant proportion (between 15% and 30%) of those in the performing arts, cultural heritage, music, publishing and print media and language industry sectors function as partnerships. In contrast, 23% of the enterprises in the visual arts sector are registered as civil law partnerships (Figure 2).

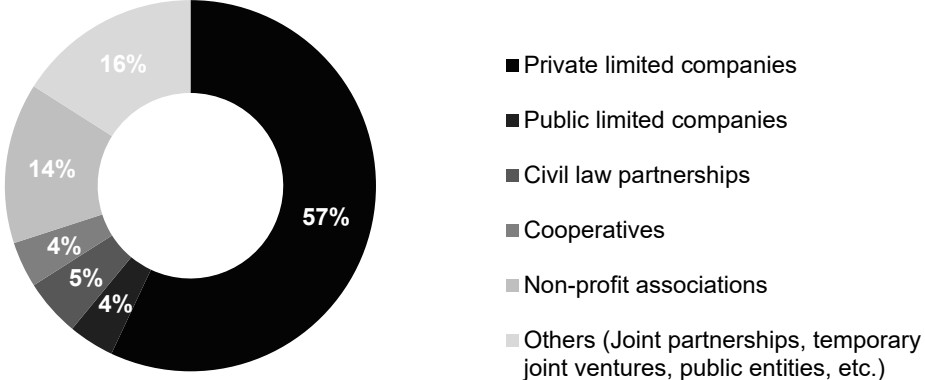

**Figure 2.** A breakdown of cultural and creative industries in the Basque Country by type of legal structure. Source: Compiled by authors on the basis of information extracted from the SABI database.

*3.2. Economic Activity*

Although the disperse archipelago of Basque cultural and creative enterprises might appear at first glance to be peripheral from an economic perspective, it nevertheless contains a cluster of companies that have a significant impact on the Basque economy.

Slightly more than 5000 enterprises pertaining to the Basque Country's cultural and creative industries collectively generated approximately 1.784 billion euros in annual operating revenue in 2015. The publishing and print media sector is responsible for almost a quarter of this figure (24.1%), the digital content sector 16.98%, the gastronomy sector 13.45%, the audiovisual sector 12.40% and advertising and marketing another 10.61%. These five sectors account for 77.54% the cultural and creative industry's total annual operating revenue.

In terms of revenue volume, 347 cultural and creative industry companies (6.85%) generated annual revenues of more than 1 million euros. Many in this group pertained to the same sectors mentioned above: 121 (34.87%) to gastronomy, 71 (20.46%) to publishing and print media, 34 (9.8%) to digital content, 32 (9.22%) to advertising and marketing and 27 (7.78) to audiovisual content. An impressive 82.13% of the companies that had passed the million-euro mark belonged to these five sectors (Figure 3).

As the databases consulted did not contain information on the economic activity of all of the more than 5000 companies identified in the Basque Country, the above data provides a highly representative but partial picture of sector operating revenues. Non-profit organizations have also been included in the analysis, since their ability to generate income is considered insofar as they are beneficiaries of grants and subsidies for projects through programs promoted by public administrations.

Given the gaps in the SABI database, we had to look elsewhere for the information we needed to properly gauge the weight of each sector as well as that of the cultural and creative industries as a whole in the Basque economy. ¿Quién es quién en la economía vasca?, a 2014 report compiled by Empresa XXI and underwritten by Banco Sabadell-Guipuzcoano, contained the information we required. According to this source, the 96 cultural and creative industry enterprises among the 3087 top ranking Basque companies had a combined turnover of 1006 million euros (Figure 4). Furthermore, the financial performance of this group had remained steady over the past few years.

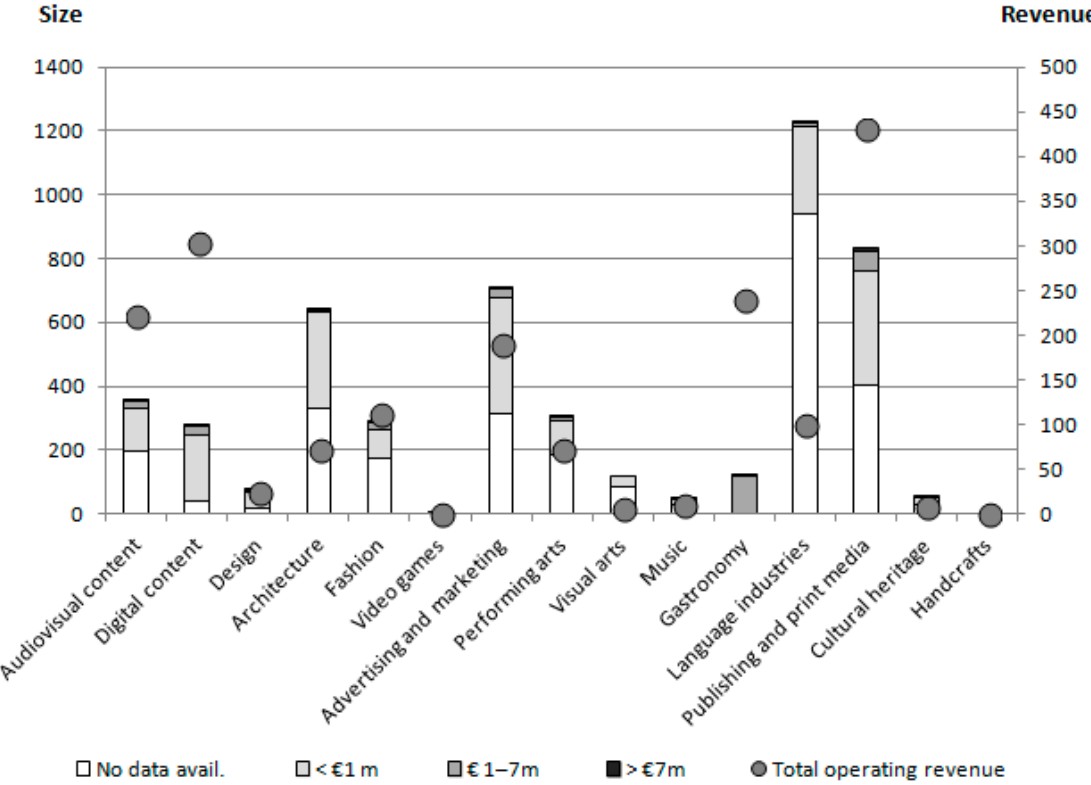

**Figure 3.** Breakdown of cultural and creative industries in the Basque Country by sector, size and operating revenue. Source: Compiled by the authors on the basis of information extracted from the SABI database.

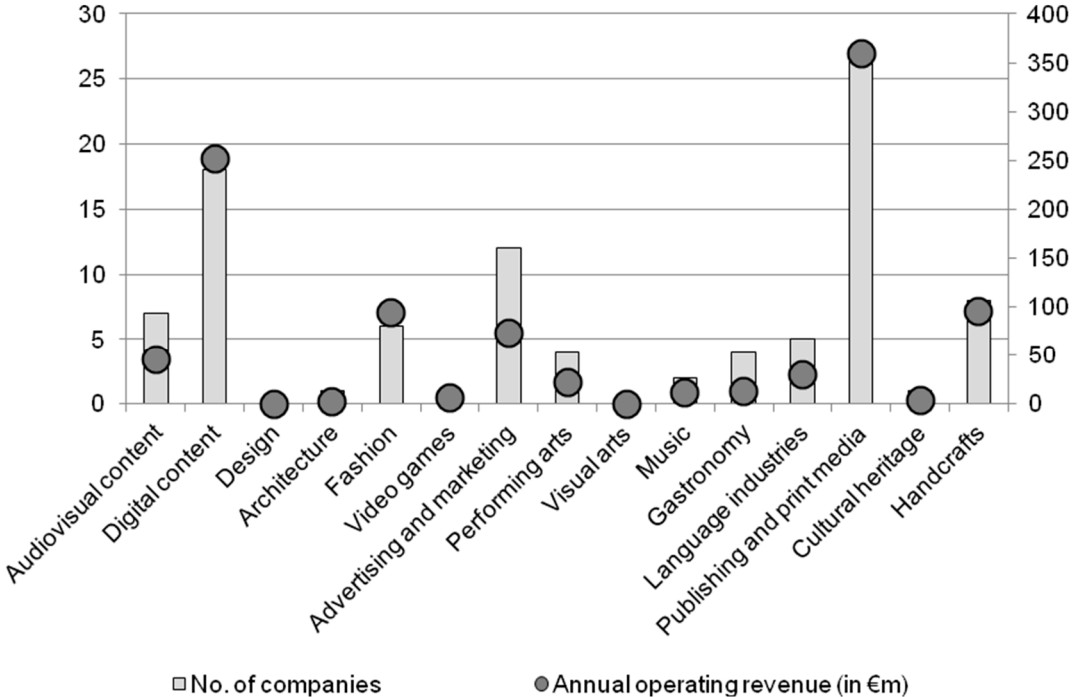

**Figure 4.** Breakdown of cultural and creative industries ranked among the Basque Country's top 3000 companies by sector, size, and annual operating revenue. Source: Compiled by the authors on the basis of Empresa XXI data.

Sales volume in the Basque Country's cultural and creative industries tends to be concentrated in certain sectors. A look at the statistics reveals that five sectors are responsible for 86.78% of cultural and creative industry's sales generated in the autonomous community: publishing and print media, digital content, handcrafts, fashion and advertising and marketing (Figure 5).

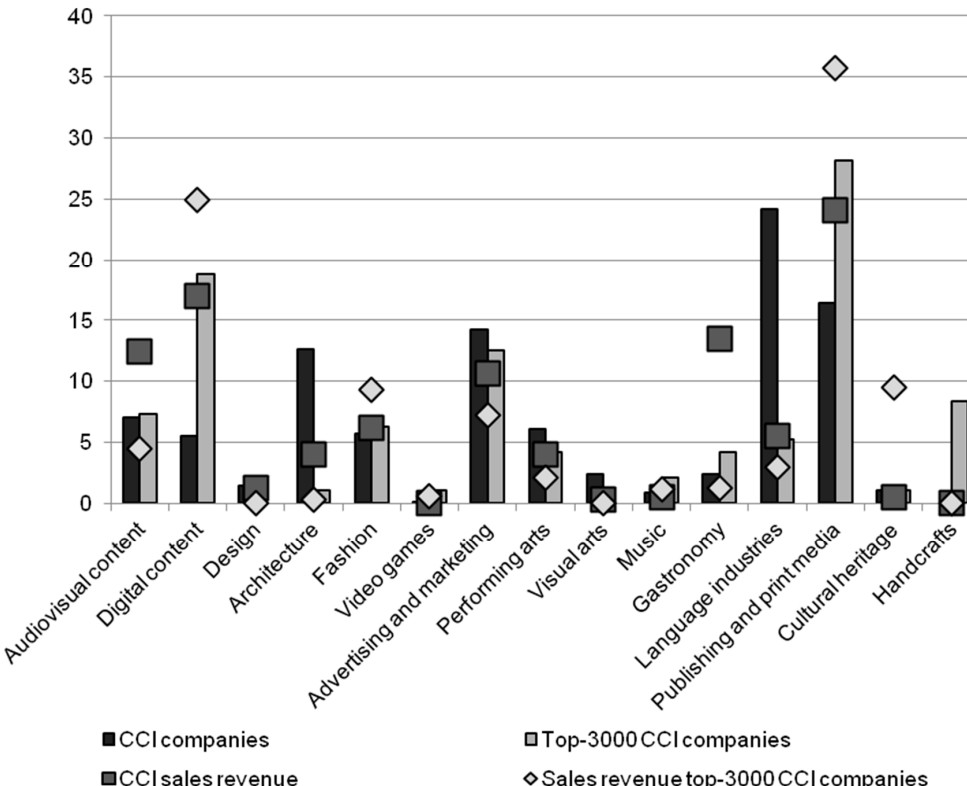

**Figure 5.** Comparison of cultural and creative industry sectors by number of enterprises and sales revenue (%). Source: Compiled by the authors on the basis of SABI database and Empresa XXI data.

In summary, the data analysed show that it is a quantitatively significant sector but with a weak business structure. Therefore, its relevance in quantitative terms should not be confused with the strength of its business structure. In this sense, it can also be affirmed that the sector is not very structured and is not very visible, since the number of activities is very high and most of the companies have a small size.

Sales volume in the Basque Country's cultural and creative industries tends to be concentrated in certain sectors. A look at the statistics reveals that five sectors are responsible for 86.78% of cultural and creative industry's sales generated in the autonomous community: publishing and print media, digital content, handcrafts, fashion and advertising and marketing (Figure 5).

In summary, the data analysed show that it is a quantitatively significant sector but with a weak business structure. Therefore, its relevance in quantitative terms should not be confused with the strength of its business structure. In this sense, it can also be affirmed that the sector is not very structured and is not very visible, since the number of activities is very high and most of the companies have a small size.

### 3.3. Employment Generated by Cultural and Creative Industries

Taking a look at the employment generated by the archipelago of businesses constituting the Basque Country's cultural and creative industries is another way of gauging the role they play in the Basque economy. Although the SABI database provided employment data for less than half of the companies in the study sample, we were able to ascertain that of the 2098 enterprises we were

able to find information on, 1479 (70.5%) had fewer than 5 employees, 203 (9.68%) had between 6 and 9, 371 (17.68%) employed between 10 and 49 people and 45 (2.14%) maintained a staff of over 50 (Figure 6). In general, companies are small in size, based on self-employment and that hardly create wage employment.

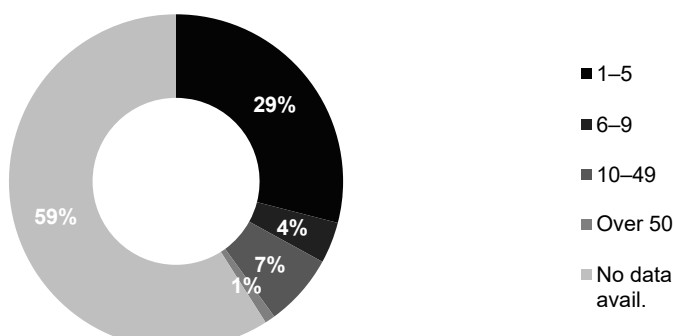

**Figure 6.** Breakdown of ICC enterprises in the Basque Country by number of employees. Source: Compiled by the authors on the basis of information extracted from the SABI database.

The number of people employed by companies in the audiovisual, digital content, cultural heritage, fashion and gastronomy sectors tended to be higher than the combined average of 7.5 employees for the cultural and creative industry as a whole.

The sector breakdown of jobs generated by the 96 enterprises included on Empresa XXI's list of the top 3000 Basque companies provided in Figures 6 and 7 highlights yet another facet of their weight within the Basque economy. These companies collectively provide 7213 jobs and the nine largest employ more than 100 people.

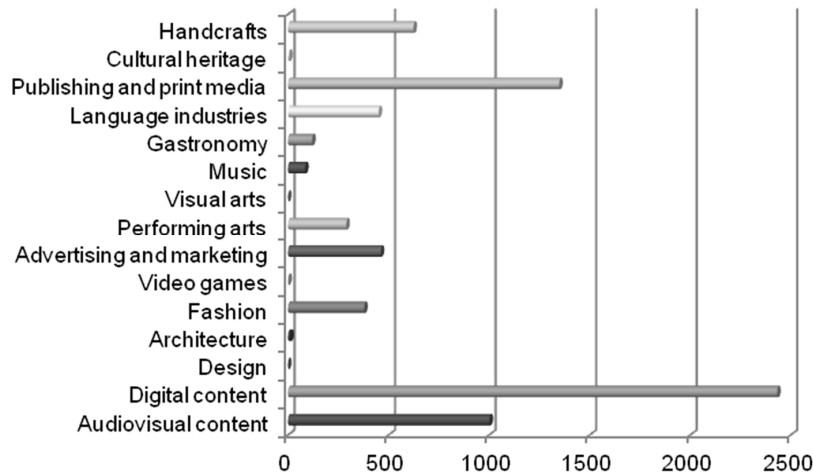

**Figure 7.** Number of people employed by Top-3000 cultural and creative industry companies. Source: Compiled by the authors on the basis of Empresa XXI data.

Employment in the Basque cultural and creative industry is concentrated in a few sectors. Digital content is responsible for 34%, publishing and print media 19% and audiovisual content a further 14%. Firms in these three areas provide two out of three jobs generated by cultural and creative industry enterprises on the Basque top-3000 list.

*3.4. Government Programmes Supporting Entrepreneurship and Creativity*

To gain a better understanding of the dynamism and financing capacity of the cultural and creative industries in the Basque Country, as well as the public sector assistance they receive, we analysed

government entrepreneurship and creativity support programmes administered at the regional and provincial level throughout the autonomous community.

We were able to identify 41 governmental entrepreneurship support programmes (some of which provide funds for innovation and internationalisation) that channelled a total of €108,901,036.23 to Basque enterprises during the period 2014–2015.

Analysis revealed that 4517 businesses, 626 or 13.86% of which pertained to the cultural and creative industries, received some sort of government funding of this type during this period (Figure 8). Companies in the digital content sector accounted for 44% of the companies to win funding—an extremely high proportion in comparison to those in other sectors such as design, which accounted for 18%, and fashion and audiovisual production, which respectively accounted for 7% and 6%. On the other end of the spectrum were enterprises in the handcrafts, performing arts, visual arts and music sectors, which together accounted for less than 1% of the cultural and creative industry companies funded.

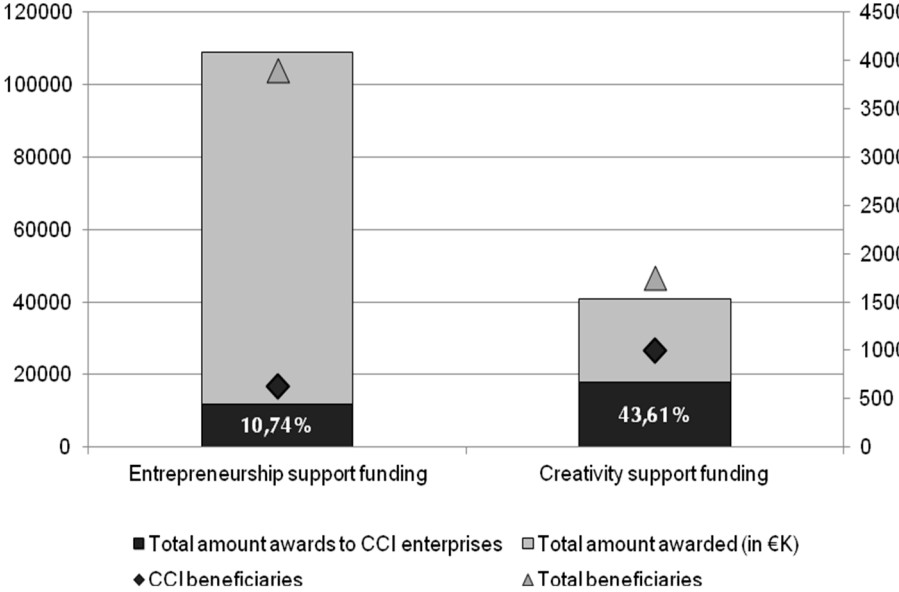

**Figure 8.** Entrepreneurship and creativity funding received by Basque cultural and creative industries. Source: Compiled by the authors on the basis of Empresa XXI data.

In terms of the monetary value of grants awarded through these programmes, cultural and creative industries received a total of €11,695.000 in direct funding (10.74% of the total amount disbursed) (Figure 9). The high proportion of that amount allotted to digital content companies—slightly over €5.6M or 48.24%—is a powerful indicator of that sector's strength in the Basque Country. Sectors on the other end of the spectrum whose enterprises received grants of less than €100,000 and as a group received less than 1% of the funding awarded were cultural heritage (0.71%), music (0.46%), visual arts (0.28%) and performing arts (0.16%). No enterprise in the handcrafts sector received funding in this category during the period examined.

An analysis of public sector support for creativity in the Basque Country indicates that a total of €42,053,316.75 was channelled to 2249 entities during the period 2014–2015 via 82 separate competitive funding processes and that 1274 (46.34%) of the beneficiaries were city or town councils or individuals, 995 (36.19%) were cultural and creative industries and 480 (17.46%) were companies falling in other categories.

If town councils and individuals are excluded from the calculation, we quickly see that cultural and creative industry enterprises accounted for an impressive 67.84% of the companies awarded funding for creativity but only 14% of those awarded funding earmarked for entrepreneurial support. Enterprises in sectors such as music and the visual and performing arts that received virtually no

funding from programmes providing entrepreneurial support fared far better in the category of creativity and figured more heavily on the lists of recipients of that type of funding programme.

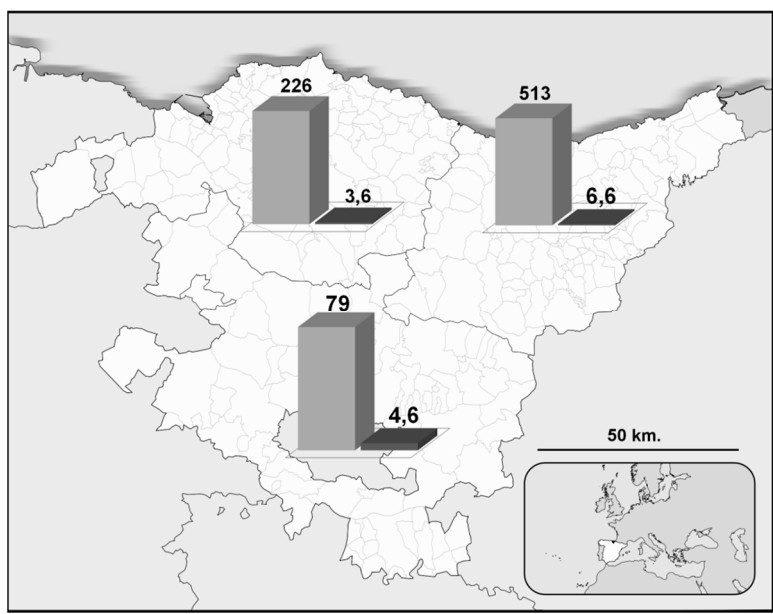

**Figure 9.** Number of cultural and creative industry companies and amount of direct funding obtained, by provinces. Source: The Authors/Miguillen.

Companies pertaining to the language, music, publishing and print media and performing arts sectors constituted 70% of the beneficiaries of public funding programmes focusing on creativity during the period examined. The sectors in which the fewest enterprises received this kind of support were handcrafts, fashion, advertising and marketing, gastronomy, architecture and video games (Figure 10).

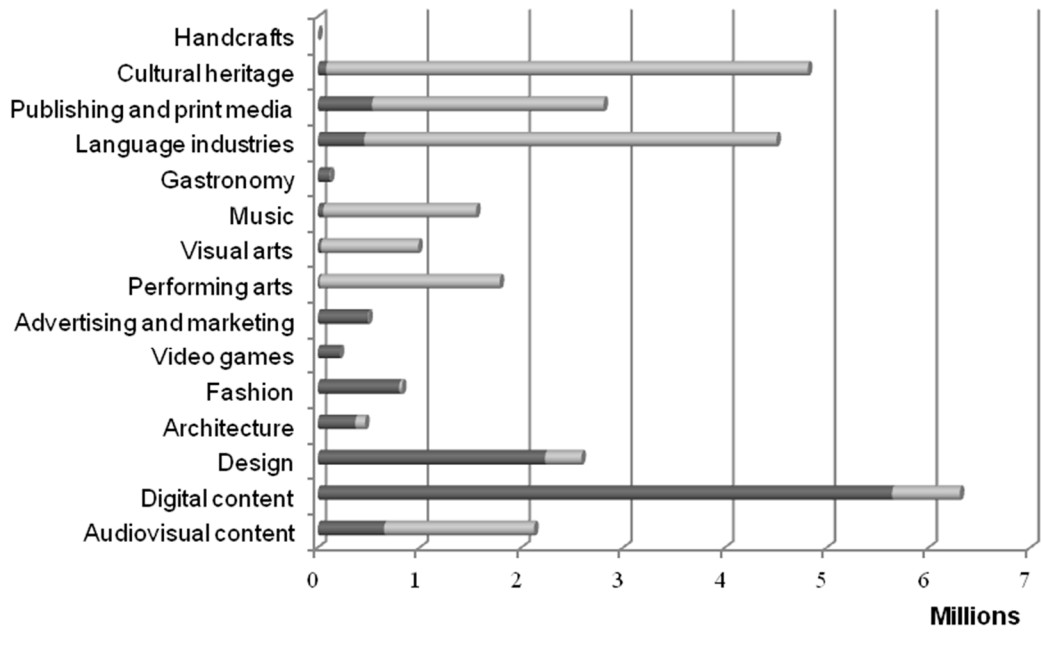

**Figure 10.** Distribution of entrepreneurial and creativity support funding awarded to Basque cultural and creative industries by sector. Source: Compiled by the authors on the basis of Empresa XXI data.

The cultural heritage sector took home the greatest share of the creativity funding pie (26.43%). The language industries ranked a close second with 22.61%, publishing and print media received 12.70% and performing arts 9.87%. The sectors receiving the least funding in this category were architecture, fashion, advertising and marketing, gastronomy, handcrafts and video games.

## 4. Discussion

The findings of this study indicate that cultural and creative industries constitute an emerging, dynamic sector of Basque economy, along the same lines as established by other research on regional level, either in Spain (Rodríguez et al. 2017; Verón-Lassa et al. 2017; Murciano and González 2018) or at international level (Boschma and Fritsch 2009; Fleischmann et al. 2017b; Correa-Quezada et al. 2018).

From an economic perspective, the financial performance of cultural and creative enterprises is the best indicator of their social and economic relevance. In this sense, the results confirm that its impact on the regional economy is clearly noticeable (Aguado 2010).

This study provides tangible evidence of the impact cultural and creative industry activities have on the Basque economy and employment market, and confirms their ability to create employment and generate economic activity, as is the case at European level (Power 2011). Our analysis revealed that 96 of the Basque Country's 3000 top performing businesses pertain to the cultural and creative industries. Collectively, these companies generate an annual sales volume of over 1 billion euros and provide a total of 7213 jobs. Such figures attest not only to these organizations' present economic viability, but their future potential as well, considering their emphasis on the three factors key to competitiveness: creativity, knowledge and technology.

However, we also find many companies that have a limited ability to generate income and difficulties to expand their business model. A sector of these characteristics tends to concentrate its activities in small productive and geographic spaces that due to their dimensions and requirements are more affordable.

These statistics fall in line with data for the country as a whole, which indicate that Spain's creative industries are responsible for 5.75% of its production (a percentage slightly below the EU average) and 6.5% of its jobs, figures that situate Spain fifth among European countries in terms of creative sector production [56].

Our examination of the distribution of public funding for entrepreneurship support indicates that some sectors within the Basque Country's cultural and creative industries are more dynamic than others in this area. At first glance, these companies appear to receive their proportional share of consideration and financial support. As a block they constitute 14% of the beneficiaries of these programmes and receive 11% of the total money awarded. A closer inspection nevertheless reveals that very few sectors get a significant share of the pie. During the two funding cycles analysed for this study, the digital content sector received almost half of the money granted to cultural and creative industries whereas others were given what could be construed as token amounts.

The opposite occurred in the case of creativity grants, a category in which sectors such as cultural heritage, publishing and print media, performing arts and the language industry that had received little or no entrepreneurship funding did very well.

## 5. Conclusions

On the basis of the data compiled and analysis performed during this study, it is fair to say that the cultural and creative industries in Basque Country constitute an archipelago of heterogeneous entities that includes both highly profitable, top-performing companies and organizations with more modest commercial ambitions. Although based on the NACE Rev.2 and CNAE-09 classifications, it can be observed that the diversity of activities within each sub-sector is very broad. This heterogeneity can also be observed in the unequal size of the subsectors, their varied corporate composition or their capacity to obtain benefits.

Generally speaking, the players that make up this ecosystem can be divided into three categories.

The first includes highly profit-oriented enterprises capable of generating substantial earned revenue that have a presence in local and international markets, a number of which play an important role in the Basque economy. Some are engaged in traditional sectors such as publishing and print media and others in emerging sectors such as digital content.

The second includes companies and other entities devoted to a particular aspect of cultural development such as cultural heritage or the performing arts whose missions and activities are less aligned with conventional business models.

The third and final covers what would best be described as hybrid organizations midway between the other two that have individual identities and approaches to cultural and creative endeavour.

In summary, in a society ever more reliant on creativity and knowledge in which cultural and creative goods and services are in increasing demand, it is possible to measure the activities of cultural and creative industries as a group in terms of the present and future value they bring to the economies in which they function.

Likewise, the development of this niche of opportunity makes it necessary to expand public-private cooperation (e.g., the laboratory for the promotion and strengthening of cultural and creative Industries created within the framework of the *Etorkizuna Eraikiz* initiative) and generate a state of trust between the promoters of public policies and the agents of the sector.

In light of the non-economic value of goods and services produced by the cultural and creative industries, these findings in no way imply that culture and creativity should be understood strictly in economic or industrial terms (Herrero 2011). Their existence neither impedes the development of other non-commercial forms of cultural and creative activity nor exempts public funders and policy makers from recognizing the value of and supporting these alternative initiatives.

These two different conceptions of cultural and creative expression and production will undoubtedly oblige public funding entities to develop a dual strategy that accommodates both of these complementary visions.

In terms of impact, the results of this research reveal certain strengths of the cultural and creative industries in Basque Country, such as the constant growth of the number of companies, the ability to generate a significant volume of employment, the growing orientation towards innovation in the creation and development of products and services and the existence of a demanding market for the consumption of culture and creative services.

In parallel, this sector also shows some weaknesses, such as the heterogeneity and diverse orientation of cultural and creative activities, the precariousness of the business structure (size and legal personality), the conditioning factors for the increase in company income, the differences between the different activity subsectors or the high dependence on public financing.

In terms of future lines of work, and based on the previous analysis, an action-oriented research may focus on some of the following activities and strategies:

1. Promote the internal structuring of the sector based on smart specialization principles, favouring clusterization initiatives.
2. Promote public-private coordination to overcome the problems of financing and economic solvency of the sector.
3. Make the sector visible by creating a battery of indicators that measure its evolution and development and giving it its own space in official statistics.

**Author Contributions:** Conceptualization, X.B.-I. and A.U.-S.; methodology, X.B.-I. and A.U.-S.; formal analysis, X.B.-I. and A.U.-S.; investigation, S.P.-F.; resources, S.P.-F.; writing—original draft preparation, S.P.-F.; writing—review and editing, A.U.-S.; supervision, X.B.-I. All authors have read and agreed to the published version of the manuscript.

**Funding:** This research received no external funding.

**Conflicts of Interest:** The authors declare no conflict of interest.

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
