# Peer review of "The Archipelago of Cultural and Creative Industries: A Case Study of the Basque Country"

_economies, doi:10.3390/economies8010021_

Round 1
Reviewer 1 Report
The paper aims to piece together an accurate picture of the size, characteristics and economic impact of the cultural and creative industries based in the Basque Country. The theoretical background is missing. Objectives are poor. I also miss international comparison. Research hypotheses must be supported by citations. It is also possible to answer hypotheses without any analyses. Results and methodology are poor. Statistical methods are missing. I propose authors to work with data and to conduct deep analyses, define research questions or hypotheses based on current knowledge and make comparison with other countries. Most of references are old. Where is the gap of this study?
Author Response
Answers to Reviewer #1: |
|
The authors would like to thank the reviewer for her valuable comments and suggestions, all of which guided our revisions to the manuscript and improved its quality. |
|
Comment (1) |
The theoretical background is missing. |
Author’s response |
The authors have completed the theoretical background, which is included in the Introduction following the journal's paper structure, with recent articles about the impact of cultural and creative industries at regional level. |
Comment (2) |
Objectives are poor |
Author’s response |
Research questions have been included at the end of the introduction to clarify the objectives of the research. |
Comment (3) |
I also miss international comparison |
Author’s response |
We agree with the reviewer, and references about similar international research have been included in order to provide some comparison possibilities
Fleischmann, K.; Daniel, R.; Welters, R. Developing a regional economy through creative industries: innovation capacity in a regional Australian city. Creative Ind. J. 2017, 10, 119-138. Wu, M; Li, Q. Impact of Cultural and Creative Industries on Regional Economic Development in China— A Spatial Econometric Approach. Res. in World Econ. 2018, 9, 46-60. Fleischmann, K; Welters, R.; Daniel, R. Creative Industries and Regional Economic Development: Can a Creative Industries Hub Spark New Ways to Grow a Regional Economy? Australas. J. of Reg. Stud. 2017, 23, 217-242. Correa-Quezada, R.; Álvarez-García, J.; Del Río-Rama, M.C. ; Maldonado-Erazo, C.P. Role of Creative Industries as a Regional Growth Factor. Sustainability 2018, 10, 1-14.
|
Comment (4) |
Research hypotheses must be supported by citations. It is also possible to answer hypotheses without any analyses |
Author’s response |
Research hypotheses are related with the following research on Introduction: H1 – The cultural and creative industries based in the Basque Country have a significant, measureable economic impact in terms of their size, productivity and employment opportunities they generate. (v.g 11,13) H2 – The Basque Country’s cultural and creative industries are diverse, in the sense that all of the sectors pertaining to them produce identifiable and measurable activity, but the distribution of business activity is uneven with the greater part corresponding to a small number of players. (v.g. 14) H3 – Cultural and creative industries in the Basque Country take advantage of government funding programmes supporting entrepreneurship and creativity administered at various levels throughout the autonomous community. (v.g. 40, 41)
We did not want to repeat the references when we understood that it was deduced from the previous section, but if it is considered necessary we will, of course.
|
Comment (5) |
Results and methodology are poor. Statistical methods are missing. |
Author’s response |
Complementary information has been added to the Material and Methods section to make it more precise. |
Comment (6) |
Most of references are old |
Author’s response |
Following reviewer's advice we have included more up-to-date references. With this change, 2/3 of the references are from the last 10 years. |
The authors will gladly answer/correct any other question that may raise.
Thank you very much for your help. |
Reviewer 2 Report
Dear Author/s,
Thank you for your paper.
You have provided interesting thoughts and statistics in your paper. However, at that moment I think that your paper is not finished yet.
You must to start with introduction and rewrite it by providing aim of the research, you research questions. Also, pls describe more research in the field as paper audience will be international and not only from Basconia. You should work on special part as literature review as now it is unclear how do you contribute to the global discussion. You must to rewrite your part on Material and Methods. Now it is unclear why do you mean that you specific objectives will help to measure an impact. Even that there are works on Bilbao Gugenheim museum that usually provides information on impact of museum to the local economy. You results section looks like description of the statistics without explanation of the impact and without providing causalities of it. In your conclusions you can provide to the readers what are the results of your research by concluding it.Author Response
Answers to Reviewer #2: |
|
Comment (0) |
Thank you for your paper. |
Author’s response |
The authors wish to thank Reviewer 2 for the helpful comments about our manuscript. |
Comment (1) |
You must to start with introduction and rewrite it by providing aim of the research, you research questions. |
Author’s response |
Research questions have been included at the end of the Introduction. |
Comment (2) |
Also, pls describe more research in the field as paper audience will be international and not only from Basconia |
Author’s response |
Following reviewer's advice, an explanation about the Basque Country has been included in the Introduction. Abstract has also been adapted. |
Comment (3) |
You should work on special part as literature review as now it is unclear how do you contribute to the global discussion. |
Author’s response |
The introduction and conclusions have been completed to align the results of this research with previous studies on the impact of cultural and creative industries at the regional level.
Aguado, L. Estadísticas culturales: Una mirada desde la economía de la cultura. Cuad. de Admin. 2010, 23, 107–141. Fleischmann, K.; Daniel, R.; Welters, R. Developing a regional economy through creative industries: innovation capacity in a regional Australian city. Creative Ind. J. 2017, 10, 119-138. Wu, M; Li, Q. Impact of Cultural and Creative Industries on Regional Economic Development in China— A Spatial Econometric Approach. Res. in World Econ. 2018, 9, 46-60. Fleischmann, K; Welters, R.; Daniel, R. Creative Industries and Regional Economic Development: Can a Creative Industries Hub Spark New Ways to Grow a Regional Economy? Australas. J. of Reg. Stud. 2017, 23, 217-242. Correa-Quezada, R.; Álvarez-García, J.; Del Río-Rama, M.C. ; Maldonado-Erazo, C.P. Role of Creative Industries as a Regional Growth Factor. Sustainability 2018, 10, 1-14.
|
Comment (4) |
You must to rewrite your part on Material and Methods. Now it is unclear why do you mean that you specific objectives will help to measure an impact. Even that there are works on Bilbao Guggenheim museum that usually provides information on impact of museum to the local economy. |
Author’s response |
Complementary information has been added to the Material and Methods section to make it more precise. As the reviewer rightly points out, the emblematic case of the Guggenheim Museum in Bilbao is one of the most prominent in the field of cultural and creative industries in Euskadi. |
Comment (5) |
You results section looks like description of the statistics without explanation of the impact and without providing causalities of it. |
Author’s response |
The reviewer is right, part of the objectives of the study are to describe the nature and characteristics of the cultural and creative sector in the Basque Country. The authors have tried to align the results of the study accordingly to the research questions. |
Comment (6) |
In your conclusions you can provide to the readers what are the results of your research by concluding it. |
Author’s response |
The discussion section has been rewritten to put into relation the results and the previous research. |
The authors will gladly answer/correct any other question that may raise.
Thank you very much for your help. |
Reviewer 3 Report
General:
1.) well written and analysed; however, the data statistically analysed are a somewhat outdated representing 2014 and 2015. Having in mind that collection of data regarding CCIs is extremely difficult (which is explained), it is advised to, at least, point why newer data could not have been gathered, if not providing them.
2.) there's a confusion on the handcrafts sector: while in the text it is referred that they were not analysed, they are still present in the analysis as well as in Figures. Either explain or correct.
3.) an explanation of language industries would be useful (maybe as a footnote): what does it actually comprise?
4.) lines 159-160 - when referring to cultural and creative industries, an explanation of the difference could be provided (maybe as a footnote): how did you conceive this difference in this study.
Figures: Figures 2 and 6 do not result clearly; they could have been cut off for the purpose of better design, however, they result as partial images, so replace them. Further on, in line 270 you are actually referring to Figure 6 and not 7, so correct it.
Language: minor corrections are required; e.g. sometimes you use "handcrafts", other times "handicrafts"; line 206 - databases and not data bases; line 194-195 "The slightly more than..." - omit "The"; line 85 - "the determine" - omit "the".
Author Response
Answers to Reviewer #3: |
|
Comment (0) |
Well written and analysed |
Author’s response |
The authors wish to thank Reviewer 3 for the helpful comments about our manuscript. |
Comment (1) |
The data statistically analysed are a somewhat outdated representing 2014 and 2015. Having in mind that collection of data regarding CCIs is extremely difficult (which is explained), it is advised to, at least, point why newer data could not have been gathered, if not providing them. |
Author’s response |
The authors share the reviewer's concern. In any case, the text collects the comparable data available at the time of writing the text. Unfortunately, some of the sources used are not updated annually. |
Comment (2) |
There's a confusion on the handcrafts sector: while in the text it is referred that they were not analysed, they are still present in the analysis as well as in Figures. Either explain or correct. |
Author’s response |
Thank you very much. It's true, the note on Table 1 could be misleading. It's been changed to "partially analysed", since we do have data from this sector from SABI and Empresa XXI data (figures 1,3,4,5,7). |
Comment (3) |
An explanation of language industries would be useful (maybe as a footnote): what does it actually comprise? |
Author’s response |
This explanation has been added: "includes translation, content -terminology, lexicography, etc.- language teaching and language technologies". It's a strong sector, since the Basque Country has two official languages (Spanish and Basque). |
Comment (4) |
Lines 159-160 - when referring to cultural and creative industries, an explanation of the difference could be provided (maybe as a footnote): how did you conceive this difference in this study. |
Author’s response |
As noted in the Introduction and the Methods section, in this text the concept of cultural and creative industries has been used as a single unit, following the NACE (Rev. 2) and CNAE-09 classifications. We are aware that other uses of this concepts have been done, but we have followed the path of other studies listed in references (v.g. 1-7) |
Comment (5) |
Figures: Figures 2 and 6 do not result clearly; they could have been cut off for the purpose of better design, however, they result as partial images, so replace them. |
Author’s response |
Both figures have been replaced and now are displayed correctly. |
Comment (6) |
Further on, in line 270 you are actually referring to Figure 6 and not 7, so correct it. |
Author’s response |
Thank you very much for the correction, the reference has been modified. |
Comment (7) |
Language: minor corrections are required; e.g. sometimes you use "handcrafts", other times "handicrafts"; line 206 - databases and not data bases; line 194-195 "The slightly more than..." - omit "The"; line 85 - "the determine" - omit "the". |
|
Handcrafts and database are the right words, the text has been revised accordingly. Sentences have been corrected. |
The authors will gladly answer/correct any other question that may raise.
Thank you very much for your help. |
Reviewer 4 Report
The contribution of the work to the discussion is not at all explained. I advise the authors to discuss the results in the light of the existing theory, namely what is discussed in the cluster theory and recent publications concerning geographical proximity and its effect on local/regional dissemination and competitiveness.
Author Response
Answers to Reviewer #4: |
|
Comment (0) |
The contribution of the work to the discussion is not at all explained. I advise the authors to discuss the results in the light of the existing theory, namely what is discussed in the cluster theory and recent publications concerning geographical proximity and its effect on local/regional dissemination and competitiveness. |
Author’s response |
The authors wish to thank Reviewer 3 for the helpful comments about our manuscript. The introduction (theoretical framework) and the discussion have been modified to introduce the suggested perspective. |
The authors will gladly answer/correct any other question that may raise.
Thank you very much for your help. |
Round 2
Reviewer 1 Report
The authors have incorporated most of the comments. I therefore agree with the publication of the article in its current form. However, in the future, I recommend authors to use more statistical models, or a combination of more research methods. Furthermore, I recommend authors to focus on international analyses. I wish the authors good luck in further publishing and research activities.
Author Response
The authors sincerely appreciate the comments of the reviewer, and will take them into account in future publications as well. Thank you very much for your help.
Reviewer 2 Report
Thank you for re-submitting. I would like that you improve you conclusions by finally stating what is the significance of your point and directions of the future research.
All the best
Author Response
Answers to Reviewer #2: |
|
The authors sincerely appreciate the comments of the reviewer, and will take them into account in future publications as well. Thank you very much for your help. |
|
Comment (1) |
I would like that you improve you conclusions by finally stating what is the significance of your point and directions of the future research. |
Author’s response |
Conclusions have been improved following reviewer's advice. We have underlined the strengths and weaknesses of the cultural and creative industries and suggested some possible future lines of work for an action oriented research (aprox. 200 words). Thank you very much. |
Reviewer 4 Report
The changes were done. Thank you.
Author Response

(The authors gave the same response as above.)
